# Efficient Recommendation Unlearning via Task Vector Arithmetic in Shared Space

## Abstract

Driven by the growing need for data privacy, machine unlearning seeks to efficiently remove the influence of specific data from trained models without costly retraining. This challenge is particularly sensitive in recommendation unlearning because collaborative filtering (CF) inherently entangles interactions' influence across the entire user-item latent space, making its precise removal non-trivial. However, prevailing paradigms exhibit fundamental limitations; partition-based methods fragment the interaction structure by design, while influence function-based approaches focus on localized parameter adjustments, failing to capture broader collaborative patterns. In this paper, we propose **COVA** (COllaborative Vector Arithmetic), a novel framework that directly address these issues. Specifically, COVA constructs shared orthogonal latent space that preserves collaborative patterns across the entire interaction matrix. Within this space, unlearning is performed by subtracting task vectors. Notably, whereas task vector arithmetic traditionally operates in the parameter space, we reinterpret it for the embedding space to align with the learning mechanism of CF. Therefore, our output-level approach operates directly on the prediction matrix of any CF model, without any access to model internals or training procedures. Experiments on three benchmark datasets demonstrate that COVA improves unlearning completeness by up to 18.83% and achieves a speedup ranging from 15 to 38.5 times over the strongest baseline, while maintaining comparable utility to the retrained model.

## 1 Introduction

The increasing demand for data privacy has established machine unlearning (Cao & Yang, 2015) as an active research area, aiming to remove specific data influence from trained models while preserving remaining knowledge (Xu et al., 2024; Qu et al., 2024; Nguyen et al., 2025). The *gold standard* is to retrain the model from scratch, but this is computationally prohibitive especially for large-scale systems. Thus, unlearning methods (Xu et al., 2023; Li et al., 2024b) focus on approximating retraining while satisfying three objectives: completeness (completely eliminating the influence of deleted data), utility (preserving model performance), and efficiency (providing a clear computational advantage over full retraining).

While these objectives are universal, collaborative filtering (CF)—the cornerstone of recommender systems (Schafer et al., 2007; Su & Khoshgoftaar, 2009)—introduces unique challenges (He et al., 2017; Dou et al., 2025). Unlike domains like NLP or CV, where data instances are independent, CF models learn from interconnected user-item interactions (Koren et al., 2009; He et al., 2020). The global distribution of a single interaction's influence, a result of jointly optimized embeddings, fundamentally distinguishes recommendation unlearning (RU) from conventional unlearning. Effective RU therefore requires both disentangling and deleting these rippling collaborative influences.

Existing approaches in RU fall primarily into two categories: partition-based (P-based) and influence function-based (IF-based). P-based methods (Bourtoule et al., 2021; Chen et al., 2022; Li et al., 2023a) enable efficient unlearning by partitioning data to retrain only affected sub-models, using strategies like similarity-based grouping to maintain collaborative signals. IF-based techniques (Wu et al., 2023a;b; Li et al., 2023b) approximate the influence of interactions to adjust model parameters directly. Recent advance (Zhang et al., 2024) expands the update scope to include neighboring embeddings.

Despite these advancements, both paradigms suffer from fundamental limitations: (1) incomplete modeling of collaborative effects and (2) computational overhead. For incomplete modeling, P-based methods deliberately partition data into isolated shards, severing collaborative signals across user-item connections (Liu et al., 2023; Li et al., 2024a), while IF-based methods remain fundamentally local even with expanded neighborhoods. For computational overhead, P-based methods incur costs comparable to full retraining when requests span multiple partitions (Chen et al., 2025), while IF-based approaches require multiple iterations without convergence guarantees.

In this paper, we propose **COVA** (COllaborative Vector Arithmetic), a novel unlearning framework for CF that addresses the limitations of prior works. To account for the collaborative signals, we employ singular value decomposition (SVD) to construct a unified latent space that preserves collaborative patterns across all user-item relationships. Within the decomposed space, we perform unlearning via task vector arithmetic (Ilharco et al., 2023; Kim et al., 2024)—a technique that enables single-step, non-iterative operation. While task vectors have demonstrated impressive efficiency (Ortiz-Jimenez et al., 2023; Zhang et al., 2025; Li et al., 2025) in NLP and CV through direct weight manipulation, CF's embedding-based learning mechanism and entangled embeddings prevent the application of task vector arithmetic. We resolve this incompatibility by adapting the core principle of task vectors—originally designed for the parameter space—to operate directly within the embedding space, bringing the computational benefits of vector operations to RU.

The core contribution of COVA is its novel formulation of a matrix-level approach on data, which doesn't require any access to model internals. Unlike the existing paradigms that require direct manipulation of user/item embeddings or model parameters, COVA operates on matrices like raw user-item interaction or prediction matrix that contain the complete collaborative landscape. This shift from interaction-level to matrix-level operations enables us to preserve and manipulate collaborative patterns rather than attempting to disentangle individual contributions. Through extensive experiments, we validate our approach across three evaluation aspects: unlearning completeness, recommendation utility, and computational efficiency, achieving substantial improvements.

We outline below for some of the general observations we achieved:

- **A New Unlearning Framework.** We introduce COVA, a novel framework that addresses the fundamental limitations of P-based and IF-based approaches.
- **Output-level Unlearning.** We develop COVA directly on model output, without requiring access to internal model parameters or training procedures.
- **Superior Performance.** Experiments demonstrate superior unlearning completeness, maintained utility comparable to retraining, and remarkable computational speedup over existing methods.

## 2 RELATED WORK

### 2.1 COLLABORATIVE FILTERING

Collaborative filtering (Sarwar et al., 2001; Schafer et al., 2007; Su & Khoshgoftaar, 2009; Parhi et al., 2017) (CF) is a widely adopted technique that learns from sparse user-item interaction matrices by embedding users and items into a shared latent space. A key characteristic of CF is the entanglement of user and item embeddings, arising from the model's capturing of both direct interactions and implicit collaborative relationships (He et al., 2017; Dou et al., 2025). This entanglement means that the influence of any single interaction becomes distributed throughout the entire representation space, as embeddings are jointly optimized through shared latent factors. Consequently, unlearning in CF presents unique challenges—simply removing an interaction from the training data is insufficient, as its influence has already been distributed across the global collaborative structure.

### 2.2 RECOMMENDATION UNLEARNING

Recommendation unlearning (RU) (Chen et al., 2024) aims to efficiently remove the influence of specific data from a trained model, but faces unique challenges due to collaborative entanglement. Although *retraining from scratch* is an ideal solution, its prohibitive computational scenario in industry has led to the development of two main paradigms: partition-based methods and influence function-based methods.

**Partition-based Methods (P-based methods)** P-based methods deliver exact unlearning by dividing the dataset into disjoint subsets and retraining only the sub-models affected by a deletion request. SISA (Bourtoule et al., 2021) pioneered the ensemble retraining approach, which RecEraser (Chen et al., 2022) later adapted specifically for RU by using user-item similarities for partitioning and attention-based aggregation. However, these approaches are constrained by two inherent limitations (Chen et al., 2025): (1) the inherent disruption of global collaborative patterns caused by data partitioning, and (2) diminished efficiency when deletion requests span multiple data subsets.

**Influence function-based Methods (IF-based methods)** IF-based methods approximate unlearning by estimating how data points affect model parameters using Hessian inverse approximations (Koh & Liang, 2017; Basu et al., 2020; Qiao et al.). SCIF (Li et al., 2023b) and IFRU (Zhang et al., 2024) exemplify this approach by adjusting embeddings of users and items associated with deleted interactions. However, these methods rely on hand-crafted rules to determine influence propagation scope, and require multiple iterations to reduce approximation errors, with computational costs escalating significantly on large-scale datasets.

## 2.3 TASK VECTOR FOR UNLEARNING

Recent studies have established task vector-based approaches as an efficient and effective unlearning paradigm (Ilharco et al., 2023; Sun et al., 2024; Kim et al., 2024; Gao et al., 2024). A task vector, defined as the parameter difference between two model states ($\tau = \theta_{\text{ft}} - \theta_{\text{pt}}$), enables unlearning through simple vector arithmetic. Specifically, $\theta_{\text{pt}}$ represents a pre-trained model and $\theta_{\text{ft}}$ represents the same model after fine-tuning on a specific task [1]. By subtracting the task vector from the fine-tuned model according to $\theta^* = \theta_{\text{ft}} - \lambda\tau$, we can effectively remove the influence of that specific task, where $\lambda$ controls the strength of the unlearning operation.

**Necessity of Reformulation in RU.** While the task vector approaches have been well established in the unlearning literature, it cannot be directly borrowed to the recommender systems. This is not due to the above mentioned unique challenge in RU, but rather due to the prevailing recommendation models, such as factorization models (Koren et al., 2009; He et al., 2017) and GNN-based approaches (Wang et al., 2019; 2020; Wu et al., 2023b). Instead of learning the weights among layers in the neural networks, these models learn the embedding of users or items through the collaborative signals. Thus, the *weight modifications* through task-vector arithmetic cannot be applied in the RU settings, which leads us to reinterpret the task vector in the embedding space. In a nutshell, we modify the learnt embeddings instead of the learnt model weights.

## 3 METHODOLOGY

In this paper, we propose **COVA** (COllaborative Vector Arithmetic), a novel framework that addresses the fundamental challenges of applying task vector concepts to recommendation unlearning (RU). Before detailing our approach, we establish the motivation for our design choices.

**Motivation.** Existing RU methods face two critical issues: insufficient consideration of collaborative relationships and computational overhead. We address these challenges by combining SVD and task vector arithmetic (Ilharco et al., 2023; Ortiz-Jimenez et al., 2023). Specifically, to resolve the issue of collaborative relationships, we leverage SVD's ability to operate on the entire interaction matrix, capturing how each interaction globally influences the latent space. Concurrently, to tackle computational overhead, we employ task vector-based operations that enable fast, non-iterative unlearning through simple linear calculations. Nevertheless, as discussed in Section 2.3, task vector arithmetic is incompatible with CF models that learn embeddings rather than parametric weights.

To overcome the incompatibility, COVA leverages a shared orthogonal space generated by SVD. By providing a unified coordinate system, this space ensures that vector arithmetic on different embedding states is mathematically coherent. Specifically, we jointly decompose three matrices: the original interaction matrix $\mathbf{Y}_{\text{original}} \in \{0, 1\}^{|\mathbb{U}| \times |\mathbb{I}|}$ containing all user-item interactions, the ideal

---

[1] In unlearning scenarios, this refers to the knowledge or data that was previously learned but now needs to be removed from the model.

matrix $\mathbf{Y}_{\text{ideal}} \in \{0, 1\}^{|\mathbb{U}| \times |\mathbb{I}|}$, which is $\mathbf{Y}_{\text{original}}$ after removing the interactions to be unlearned, and the prediction matrix $\mathbf{R}_{\text{pred}} \in \mathbb{R}^{|\mathbb{U}| \times |\mathbb{I}|}$ containing scores from a CF model trained on $\mathbf{Y}_{\text{original}}$. Here, $\mathbb{U}$ and $\mathbb{I}$ represent the sets of all users and items, respectively, with their sizes denoted by $|\mathbb{U}|$ and $|\mathbb{I}|$. The comparison between $\mathbf{Y}_{\text{original}}$ and $\mathbf{Y}_{\text{ideal}}$ identifies what to remove, while $\mathbf{R}_{\text{pred}}$ captures the learned collaborative effects requiring unlearning. By mapping these three matrices to a shared space, we perform vector arithmetic on embeddings—modifying learned user/item representations rather than model weights. Appendix A presents further discussion on alternative configurations.

## 3.1 Constructing the Shared Space

We now detail the construction of the unified matrix at the core of COVA. This matrix is formed by vertically concatenating the three matrices defined above:

$$\mathbf{A} = \begin{bmatrix} \mathbf{Y}_{\text{original}} \\ \mathbf{Y}_{\text{ideal}} \\ \mathbf{R}_{\text{pred}} \end{bmatrix} \in \mathbb{R}^{3|\mathbb{U}| \times |\mathbb{I}|}, \tag{1}$$

where $\mathbf{R}_{\text{pred}}$ is obtained from a standard CF model (e.g., MF (Koren et al., 2009) or LightGCN (He et al., 2020)) trained on $\mathbf{Y}_{\text{original}}$.

The next step is to decompose the unified matrix $\mathbf{A}$ via SVD to establish the shared latent space. However, a critical challenge arises from scale heterogeneity between the binary matrices ($\mathbf{Y}_{\text{original}}$, $\mathbf{Y}_{\text{ideal}}$) and the continuous prediction matrix ($\mathbf{R}_{\text{pred}}$). SVD inherently prioritizes reconstruction of larger-valued patterns, potentially causing it to focus predominantly on $\mathbf{R}_{\text{pred}}$ while neglecting the binary matrices—a problem that would prevent meaningful task vector computation. To ensure balanced representation of all three states, we normalize the binary matrices using user-specific statistics derived from $\mathbf{R}_{\text{pred}}$.

Let $\hat{r}_{ui}$ denote the predicted score for user $u$ and item $i$ in $\mathbf{R}_{\text{pred}}$. For users with positive interactions $\mathbb{Y}_u^+ = \{i : y_{ui} = 1\}$ and unobserved interactions $\mathbb{Y}_u^0 = \{i : y_{ui} = 0\}$, the corresponding entries in $\mathbf{Y}_{\text{original}}$, we normalize the binary matrices to match the scale of $\mathbf{R}_{\text{pred}}$. Specifically, we replace entries in $\mathbf{Y}_{\text{original}}$ and $\mathbf{Y}_{\text{ideal}}$ with user-specific statistics; positive interactions become $\bar{r}_u^+ = \frac{1}{|\mathbb{Y}_u^+|} \sum_{i \in \mathbb{Y}_u^+} \hat{r}_{ui}$, and non-interactions become $\bar{r}_u^0 = \frac{1}{|\mathbb{Y}_u^0|} \sum_{i \in \mathbb{Y}_u^0} \hat{r}_{ui}$. For interactions to be unlearned, $\mathbb{Y}_u^-$, the corresponding entries in $\mathbf{Y}_{\text{ideal}}$, we assign $\min_{i \in \mathbb{Y}_u} \hat{r}_{ui}$ to maximize separation from retained interactions while avoiding outlier sensitivity. Section 4.3.1 empirically validates this design choice against alternatives.

With the transformation and Equation 1, we reconstruct the unified matrix $\mathbf{A}$ with normalization:

$$\tilde{\mathbf{A}} = \begin{bmatrix} \tilde{\mathbf{Y}}_{\text{original}} \\ \tilde{\mathbf{Y}}_{\text{ideal}} \\ \mathbf{R}_{\text{pred}} \end{bmatrix} \in \mathbb{R}^{3|\mathbb{U}| \times |\mathbb{I}|}, \tag{2}$$

where $\tilde{\mathbf{Y}}_{\text{original}} \in \mathbb{R}^{|\mathbb{U}| \times |\mathbb{I}|}$ and $\tilde{\mathbf{Y}}_{\text{ideal}} \in \mathbb{R}^{|\mathbb{U}| \times |\mathbb{I}|}$ are the normalized versions of $\mathbf{Y}_{\text{original}}$ and $\mathbf{Y}_{\text{ideal}}$, respectively.

Although normalization ensures balanced contributions, decomposing the unified matrix $\tilde{\mathbf{A}} \in \mathbb{R}^{3|\mathbb{U}| \times |\mathbb{I}|}$ presents a scalability bottleneck. Standard full SVD would require constructing a $(3|\mathbb{U}|) \times (3|\mathbb{U}|)$ matrix with prohibitive memory complexity $O((3|\mathbb{U}|)^2)$. To address this, we adopt randomized low-rank SVD (Halko et al., 2011), which approximates a low-dimensional subspace that captures most structural properties, performing decomposition on a smaller matrix ($O(3|\mathbb{U}| \times k)$ where $k \ll 3|\mathbb{U}|$). However, even with low-rank SVD, $\tilde{\mathbf{A}} \in \mathbb{R}^{3|\mathbb{U}| \times |\mathbb{I}|}$ remains memory-intensive for large datasets. To bypass this constraint, we implement chunk-based operations dividing matrices into $c$ chunks for sequential processing, enabling datasets that exceed GPU memory. Detailed implementation is described in Appendix B.

Following this memory-efficient strategy, we compute:

$$\tilde{\mathbf{A}} \approx \mathbf{U}\mathbf{\Sigma}\mathbf{V}^\top, \quad \mathbf{U} \in \mathbb{R}^{3|\mathbb{U}| \times k}, \mathbf{\Sigma} \in \mathbb{R}^{k \times k}, \mathbf{V} \in \mathbb{R}^{|\mathbb{I}| \times k}, \tag{3}$$

where $\mathbf{U}$ and $\mathbf{V}$ are orthonormal matrices, $\mathbf{\Sigma}$ is a diagonal matrix containing $k$ singular values. By partitioning $\mathbf{U}$ into three equal blocks along the row dimension, we obtain $\mathbf{U}_{\text{original}}, \mathbf{U}_{\text{ideal}}$, and $\mathbf{U}_{\text{pred}} \in \mathbb{R}^{|\mathbb{U}| \times k}$—user embeddings for each interaction state that share the same item basis $\mathbf{V}$.

## 3.2 UNLEARNING VIA TASK VECTOR

Having established the shared orthogonal space through SVD decomposition, we perform unlearning through task vector arithmetic (Ilharco et al., 2023; Gao et al., 2024). Our approach divides the unlearning problem into two distinct components: removing the structural presence of the interactions to be unlearned and deleting their learned influence distributed through CF.

To achieve this, we leverage three embedding states from our SVD decomposition. The state $\mathbf{U}_{\text{ideal}}$ represents embeddings derived from the interaction matrix without removal targets. The state $\mathbf{U}_{\text{original}}$ includes the interactions to be unlearned in the data structure. The state $\mathbf{U}_{\text{pred}}$ captures the model's learned embeddings after training on $\mathbf{Y}_{\text{original}}$, where collaborative signals have been distributed throughout the embedding space during optimization.

With these states defined, we first capture the structural difference in the data:

$$\Delta_1 = \mathbf{U}_{\text{original}} - \mathbf{U}_{\text{ideal}}, \tag{4}$$

where $\Delta_1$ represents how the presence of the interactions to be deleted directly affects the decomposed embedding structure. However, $\Delta_1$ alone is insufficient because CF models distribute interaction influences across the entire embedding space during training.

The second vector captures these distributed effects:

$$\Delta_2 = \mathbf{U}_{\text{pred}} - \mathbf{U}_{\text{original}}. \tag{5}$$

Here, $\Delta_2$ represents the transformation from the raw interaction structure to the trained model state, including collaborative patterns that extend beyond direct data differences.

Complete unlearning requires addressing both contribution types simultaneously:

$$\mathbf{U}_{\text{unlearned}} = \mathbf{U}_{\text{pred}} - \alpha \cdot \Delta_1 - \beta \cdot \Delta_2, \tag{6}$$

$$\hat{\mathbf{R}}_{\text{unlearned}} = \mathbf{U}_{\text{unlearned}} \mathbf{\Sigma} \mathbf{V}^\top, \tag{7}$$

where $\alpha$ and $\beta$ control the strength of structural and collaborative unlearning, respectively.

**Discussion.** COVA enables tractable unlearning by separating the direct interaction differences ($\Delta_1$) from the indirect collaborative patterns learned during optimization ($\Delta_2$). The vector $\Delta_1$ captures the explicit gap between having and not having removal targets in the embedding structure, while $\Delta_2$ represents the indirect influences that emerge from CF's optimization process. Without addressing both direct and indirect components, the learned influence of removed interactions would persist throughout the embedding space, leading to incomplete unlearning. Section 4.3.2 empirically validates the necessity of both components for comprehensive unlearning.

## 4 EVALUATION

In this section, we evaluate COVA in several aspects: ❶ Completeness and Utility, ❷ Influence of Collaborative Effect, ❸ Efficiency, and ❹ Robustness.

### 4.1 EXPERIMENTS SETUP

**Data Preprocessing.** We evaluate our approach on three collaborative filtering (CF) benchmarks: Yelp2018 (Wang et al., 2019), Gowalla (Cho et al., 2011), and Amazon-Book (Ni et al., 2019). Following established protocols (He et al., 2020), we convert explicit ratings to implicit feedback and apply 5-core filtering to ensure sufficient interaction density. The dataset is partitioned into training, validation, and test sets with a 6:2:2 ratio. For conducting unlearning scenario, we randomly sample 1% of training interactions as the forget set (Zhang et al., 2024). Importantly, we employ fixed random seeds across all experiments to ensure identical forget/retain partitions for all baselines, enabling a fair comparison. Detailed preprocessing parameters are provided in Appendix C.1.

**Evaluation Metrics.** To evaluate recommendation utility after unlearning, we adopt the standard metrics Recall@20 (hereafter R@20) and NDCG@20 (N@20), which capture complementary aspects of ranking quality: retrieval effectiveness and ranking precision, respectively. For evaluating

Table 1: **Performance comparison of recommendation unlearning methods across MF and LightGCN backbones.** Metrics include Recall@20 (R@20) and NDCG@20 (N@20) for utility, and Unlearning Ratio (UR↑, percentage of forgotten items with decreased rankings) and Average Ranking Drop (ARD↓, magnitude of ranking decrease) for completeness. **Bold** indicates best performance, underlined indicates second-best. All improvements are statistically significant ($p < 0.05$).

| Backbone | Methods | Yelp2018 | | | | Gowalla | | | | Amazon-Book | | | |
|---|---|---|---|---|---|---|---|---|---|---|---|---|---|
| | | R@20 | N@20 | UR↑ | ARD↓ | R@20 | N@20 | UR↑ | ARD↓ | R@20 | N@20 | UR↑ | ARD↓ |
| MF | Original | 0.0600 | 0.0468 | - | - | 0.1281 | 0.0911 | - | - | 0.0529 | 0.0412 | - | - |
| | Retrain | 0.0600 | 0.0468 | 0.8408 | -13.8632 | 0.1282 | 0.0906 | 0.8277 | -13.5637 | 0.0529 | 0.0412 | 0.8695 | -46.1725 |
| | SISA | 0.0296 | 0.0230 | 0.5291 | -2.2327 | 0.0372 | 0.0281 | 0.4941 | -1.0976 | 0.0150 | 0.0126 | 0.5655 | -22.1065 |
| | RecEraser | 0.0427 | 0.0329 | 0.5189 | -1.8813 | 0.0801 | 0.0530 | 0.5358 | -2.9184 | 0.0369 | 0.0291 | 0.6338 | -20.4532 |
| | SCIF | 0.0547 | 0.0423 | 0.8665 | -9.6503 | 0.1110 | 0.0776 | 0.8768 | **-14.6050** | 0.0482 | 0.0376 | 0.8975 | -33.1377 |
| | IFRU | 0.0594 | 0.0461 | 0.9022 | -14.4583 | 0.1278 | 0.0908 | 0.8626 | -12.4483 | 0.0522 | 0.0408 | 0.9201 | -41.0872 |
| | COVA | **0.0601** | **0.0468** | **0.9533** | -22.1413 | 0.1279 | 0.0911 | 0.8793 | -13.4686 | 0.0523 | 0.0408 | 0.9568 | -41.7162 |
| | *Improv.* | *+1.18%* | *+1.52%* | *+5.66%* | *+53.06%* | *+0.08%* | *+0.33%* | *+0.29%* | *-7.78%* | *+0.19%* | *+0.00%* | *+3.99%* | *+1.53%* |
| LightGCN | Original | 0.0645 | 0.0505 | - | - | 0.1351 | 0.0968 | - | - | 0.0596 | 0.0466 | - | - |
| | Retrain | 0.0642 | 0.0498 | 0.7668 | -10.9370 | 0.1351 | 0.0965 | 0.7567 | -12.0339 | 0.0593 | 0.0463 | 0.8465 | -41.5210 |
| | SISA | 0.0365 | 0.0287 | 0.5385 | -2.5703 | 0.0548 | 0.0413 | 0.5411 | -3.2779 | 0.0200 | 0.0167 | 0.5553 | -11.0034 |
| | RecEraser | 0.0404 | 0.0313 | 0.5212 | -1.7056 | 0.0813 | 0.0567 | 0.5100 | -1.4596 | 0.0316 | 0.0250 | 0.5966 | -16.5469 |
| | SCIF | 0.0579 | 0.0449 | 0.7523 | -7.7871 | 0.1252 | 0.0897 | 0.8096 | -10.6624 | 0.0564 | 0.0438 | 0.8145 | -22.2425 |
| | IFRU | 0.0631 | 0.0492 | 0.7819 | -8.7764 | 0.1337 | 0.0954 | 0.7640 | -10.1190 | 0.0585 | 0.0457 | 0.8326 | -35.3207 |
| | COVA | **0.0639** | **0.0498** | **0.9291** | -17.3826 | 0.1345 | 0.0960 | 0.9168 | -16.7541 | 0.0590 | 0.0459 | 0.9596 | -39.1759 |
| | *Improv.* | *1.27%* | *1.22%* | *18.83%* | *98.05%* | *+0.60%* | *+0.63%* | *+13.24%* | *+57.10%* | *+0.85%* | *+0.44%* | *+15.25%* | *+10.88%* |

unlearning completeness, prior studies (Wu et al., 2023a;b; Chen et al., 2025) employ membership inference-based metrics (Olatunji et al., 2021; Jagielski et al.). However, these approaches rely on prediction probabilities or scores, making them unsuitable for the ranking-based nature of recommender systems. Metrics based on prediction scores, for instance, can be misleading; since different models produce scores on widely varying scales, a change in score does not guarantee a meaningful change in the final, user-facing ranked list. To address this fundamental mismatch, we propose two novel ranking-aware metrics that directly quantify the functional impact of unlearning on recommendation rankings. The first, *Unlearning Ratio (UR)*, quantifies the breadth of the effect by measuring the percentage of forgotten interactions whose rankings decrease. The second, *Average Ranking Drop (ARD)*, captures the depth of unlearning by quantifying the average magnitude of ranking degradation for affected items. Together, these complementary metrics provide an interpretable and robust evaluation framework specifically tailored for recommendation unlearning, focusing on both the coverage and intensity of ranking changes rather than abstract score variations.

**Baselines.** We evaluate the unlearning performance on two representative recommendation models: Matrix Factorization (MF) (Koren et al., 2009) and LightGCN (He et al., 2020). These two backbone architectures have always been fundamental benchmarks across recommendation unlearning studies, representing classical matrix decomposition and modern graph-based approaches respectively. We compare our proposed method with representative baselines, which can be categorized into two main groups. For partition-based methods, we compare with SISA (Bourtoule et al., 2021) and RecEraser (Chen et al., 2022). For influence function-based approaches, we adopt SCIF (Li et al., 2023b) and IFRU (Zhang et al., 2024). Detailed descriptions of each baseline are provided in Appendix C.2.

**Implementation Details.** We adopt the Bayesian Personalized Ranking (BPR) loss (Rendle et al., 2009) to align with the ranking-oriented nature of the task. All baseline models are implemented with 48-dimensional embeddings and optimized using the Adam optimizer with an early stopping criterion, following the hyperparameter configurations from the IFRU study (Zhang et al., 2024). For our proposed model, COVA, the key hyperparameters $\alpha$ and $\beta$ are tuned via a grid search over the ranges $\{5, 10, ..., 40\}$ with increment of 5 and $\{0.1, 0.2, ..., 0.9\}$ with increment of 0.1, respectively. Full experimental details and tuning results are available in Appendix C.3 and C.4. Our source code is provided for reproducibility purposes.[2]

## 4.2 RESULT AND ANALYSIS

**Completeness and Utility.** Table 1 demonstrates that COVA achieves superior unlearning completeness while maintaining model utility comparable to the gold-standard Retrain approach. For un-

---

[2]Anonymized code is available at: `https://anonymous.4open.science/r/COVA-7F64`

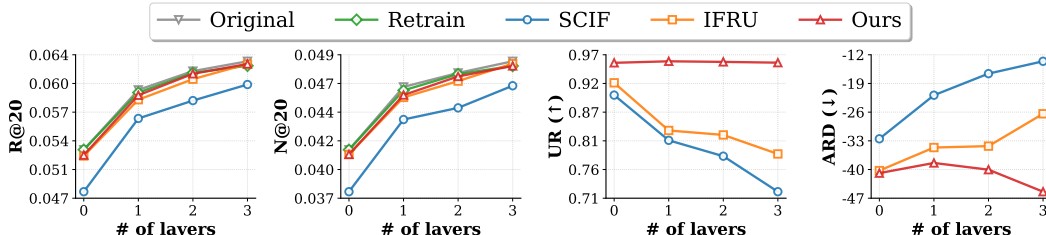

Figure 1: **Unlearning performance on Amazon-Book dataset with varying layer depths.** While deeper layers enhance collaborative signal capture and improve Recall/NDCG, existing methods (SCIF, IFRU) suffer from increased user-item entanglement. Our method maintains consistent unlearning effectiveness regardless of layer depth. The 0-layer setting corresponds to MF model.

learning completeness, COVA consistently achieves the highest UR scores across all settings, with particularly striking improvements on LightGCN—18.83% over IFRU on Yelp2018 and 13.24% on Gowalla. For ARD, COVA shows substantial improvements in most cases, such as 98.05% on Yelp2018-LightGCN. While SCIF achieves marginally better ARD on Gowalla-MF, this comes at significant utility cost—COVA maintains R@20 at 0.1279 compared to SCIF's 0.1110, a 15.2% improvement. Notably, COVA has even outperformed the Retrain model in terms of unlearning completeness in several cases. We attribute this to the potential of $\Delta_2$ to denoise the model by subtracting noisy collaborative signals introduced during optimization—a hypothesis supported by our ablation study (see Section 4.3.2). This results in an unlearned model with a more robust latent structure than is possible with simple retraining. Consequently, COVA achieves a higher degree of completeness while matching the utility of the gold-standard model, fulfilling the core objectives of unlearning.

**Collaborative Effect.** Our findings reveal a significant disparity in COVA's ability across different model architectures. For instance, on MF, COVA achieves a modest 3.33% average improvement in UR over the second-best baseline, but on LightGCN, the improvement jumps to 14.74%. The explanation lies in the architectural differences. Graph-based models like LightGCN propagate information through message passing, creating dense collaborative signals that entangle user-item interactions across multiple hops. We hypothesize that this deep entanglement of collaborative signals, exacerbated by increased layer depth, complicates the targeted removal of specific interactions. To validate this hypothesis, we analyze unlearning performance across varying GCN layer depths.

Figure 1 highlights COVA's robustness to GCN layer depth on Amazon-Book dataset. While deeper networks improve recommendation metrics (R@20, N@20) for all methods, their unlearning capabilities diverge sharply. Baselines like SCIF and IFRU show severe degradation in UR as layers increase to 3. In contrast, COVA maintains stable UR scores and even strengthens ARD on Amazon-Book dataset, demonstrating its ability to leverage, rather than be hindered by, deep collaborative signals. Further analysis and the complete results for all datasets are provided in Appendix D, which confirms that similar trends hold across all datasets. The robustness to network depth makes our approach particularly suited for real-world systems where both recommendation quality and unlearning completeness are essential requirements.

**Efficiency.** COVA inherits the computational advantage of direct parameter manipulation from task arithmetic methods—enabling single-step unlearning without iterative optimization. Table 2 benchmarks COVA's runtime against several baselines. Compared to IFRU (Zhang et al., 2024)—the strongest baseline in terms of unlearning performance—COVA achieves 6.5-10.6× speedup on the MF

Table 2: **Running time comparison.**

| Datasets | Backbone | Running Time (s) | | | | |
|---|---|---|---|---|---|---|
| | | Original | Retrain | SCIF | IFRU | COVA |
| Yelp2018 | MF | 3379 | 3324 | 715 | 3832 | **406** (-89.4%) |
| | LightGCN | 7051 | 7554 | 2443 | 10153 | **342** (-96.6%) |
| Gowalla | MF | 3471 | 3575 | 1669 | 2830 | **432** (-84.7%) |
| | LightGCN | 6342 | 6656 | 3141 | 6808 | **453** (-93.3%) |
| Amazon-Book | MF | 12135 | 11171 | 4877 | 16241 | **1531** (-91.3%) |
| | LightGCN | 23466 | 23982 | 22792 | 61240 | **1592** (-97.4%) |

backbone and 15.0-38.5× speedup on the LightGCN backbone. This improvement stems from the architectural differences. While IFRU approximates Hessian matrices to reduce computational com-

Table 3: **Memory-time trade-off analysis for our chunked implementation on LightGCN backbone. Bold** entries indicate configurations balancing memory efficiency with acceptable runtime overhead.

| Datasets | Type | SCIF | IFRU | COVA with # of chunks | | | | | | | |
| --- | --- | --- | --- | --- | --- | --- | --- | --- | --- | --- | --- |
| | | | | 1 | 2 | 5 | 10 | 15 | 20 | 25 | 30 |
| Yelp2018 | Memory (MB) | 1420 | 1782 | 5036 | 2502 | **1122** | 662 | 510 | 432 | 386 | 356 |
| | Running Time (s) | 2443 | 10153 | 342 | 544 | **713** | 856 | 911 | 1020 | 1026 | 1363 |
| Gowalla | Memory (MB) | 950 | 1150 | 5476 | 2526 | 1126 | **660** | 504 | 426 | 380 | 384 |
| | Running Time (s) | 3141 | 6808 | 453 | 538 | 708 | **854** | 888 | 994 | 1057 | 1345 |
| Amazon-Book | Memory (MB) | 2704 | 3383 | 19710 | 9542 | 4024 | **2184** | 1572 | 1266 | 1080 | 958 |
| | Running Time (s) | 22792 | 61240 | 1592 | 1657 | 2031 | **2523** | 2860 | 3013 | 3258 | 3728 |

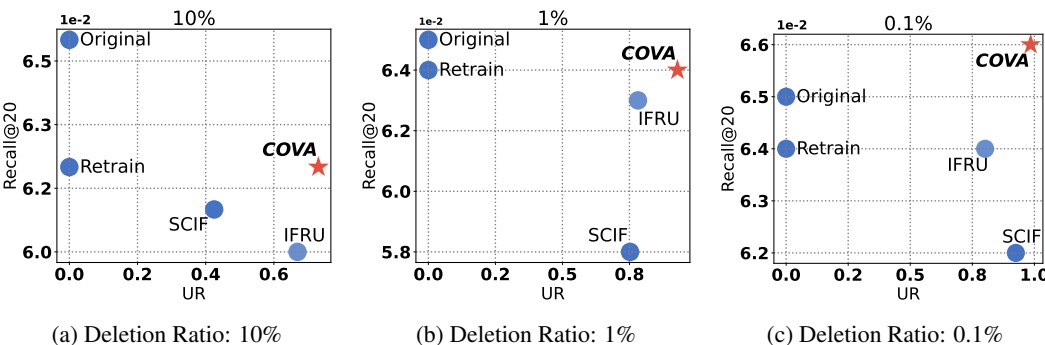

(a) Deletion Ratio: 10%     (b) Deletion Ratio: 1%     (c) Deletion Ratio: 0.1%

Figure 2: **Performance comparison of model utility and unlearning completeness on LightGCN backbone.** The deletion ratios—(a) 10%, (b) 1%, and (c) 0.1%—represent the proportion of data randomly sampled from the original training set for unlearning. Our proposed method consistently outperforms the strong baselines on both metrics across all tested scenarios.

plexity, it still requires iterative updates that often take longer than retraining due to convergence difficulties. In contrast, COVA performs unlearning through direct vector arithmetic, circumventing the limitation of slow convergence.

Additionally, Table 3 presents the memory-time trade-off analysis of our chunked implementation. While non-chunked implementation achieves the fastest runtime, it requires substantial memory—up to 19.7GB for Amazon-Book. Through partitioned matrix processing, we achieve dramatic memory reduction with acceptable runtime overhead. For instance, with 5 chunks on Yelp2018, COVA achieves 4.5× memory efficiency while the time efficiency only decreases by a factor of 2.1. The optimal configuration varies by dataset size; smaller datasets work well with 5-10 chunks, while larger datasets like Amazon-Book benefit from 10 chunks, achieving 9× memory reduction with 1.6× runtime increase. Importantly, even with chunking overhead, our method remains significantly faster than iterative baselines. The slowest configuration of COVA is still 1.8× faster than SCIF and 7.5× faster than IFRU. This flexibility allows deployment in diverse hardware settings with minimal extra time needed.

**Robustness.** To evaluate the robustness of COVA, we compare its model utility and unlearning completeness across various data deletion ratios for unlearning. We simulate a range of scenarios, from high to more realistic low ratios, with the results illustrated in Figure 2. As expected, at a high deletion ratio (e.g., 10%), all methods show a noticeable decline in model utility, struggling with the large volume of unlearning requests. Conversely, as the deletion ratio decreases to more practical levels, our method consistently achieves superior performance in both model utility and unlearning completeness compared to the baselines. Notably, in the case of the lowest ratio at 0.1% (see Figure 2c), COVA delivers the highest unlearning performance while also exhibiting greater utility than the original model. This suggests that our proposed $\Delta_2$ component is not only a facilitator of the unlearning process but also a regularizer, mitigating the model's inherent noise.

Table 4: **Ablation study on normalization strategies for COVA.** *Extreme* shows instability (Gowalla R@20: 0.1295), while our approach achieves consistently utility across all datasets.

| Methods | Yelp2018 | | Gowalla | | Amazon-Book | |
|---|---|---|---|---|---|---|
| | R@20 | UR↑ | R@20 | UR↑ | R@20 | UR↑ |
| Binary | 0.0646 | 0.4141 | 0.1353 | 0.4479 | 0.0597 | 0.6248 |
| Ternary | 0.0646 | 0.5955 | 0.1352 | 0.6174 | 0.0596 | 0.7814 |
| Statistical | 0.0647 | 0.8283 | 0.1357 | 0.7713 | 0.0595 | 0.9109 |
| Extreme | 0.0639 | 0.9306 | 0.1295 | 0.9084 | 0.0590 | 0.9605 |
| COVA | 0.0639 | 0.9291 | 0.1345 | 0.9168 | 0.0590 | 0.9596 |

Table 5: **Ablation study on task vector components:** $\Delta_1$ and $\Delta_2$. **Bold** and *italic* indicate best and worst performance among ablated configurations, respectively. Results confirm that $\Delta_1$ and $\Delta_2$ are complementary and both essential.

| Configuration | Yelp2018 | | Gowalla | | Amazon-Book | |
|---|---|---|---|---|---|---|
| | R@20 | UR↑ | R@20 | UR↑ | R@20 | UR↑ |
| w/o $\Delta_1$ | **0.0665** | 0.3912 | **0.1371** | 0.3084 | **0.0605** | 0.3251 |
| w/o $\Delta_2$ | *0.0639* | **0.7175** | *0.1353* | **0.6105** | *0.0595* | **0.7528** |
| w/o both | 0.0648 | *0.1772* | 0.1356 | *0.1393* | 0.0596 | *0.1826* |
| COVA | 0.0639 | 0.9291 | 0.1345 | 0.9168 | 0.0590 | 0.9596 |

## 4.3 ABLATION STUDY

### 4.3.1 NORMALIZATION STRATEGY FOR COVA

To validate our normalization design for addressing scale heterogeneity in COVA, we compare five normalization variants in Table 4. The *Binary* approach assigns 1 to positive interactions and 0 to both unobserved and removal targets, while *Ternary* assigns 1, 0, and -1 respectively. Both achieve poor unlearning performance despite maintaining high R@20, confirming that SVD requires proper scale differentiation. The *Statistical* approach uses user-specific statistics ($\mu_u + \sigma_u$ for positive, $\mu_u$ for unobserved, $\mu_u - \sigma_u$ for interactions to be unlearned) but shows limited unlearning effectiveness. The *Extreme* variant assigns maximum/mean/minimum values respectively, achieving the highest UR but with up to 4.3% utility degradation compared to COVA. Our approach assigns mean positive predictions to positive interactions, mean unobserved predictions to unobserved, and minimum prediction to interactions to be deleted, achieving consistently high unlearning completeness while maintaining stable utility. The *Extreme* variant's instability stems from its use of a single, distorting maximum value for positive interactions, whereas our approach uses a more conservative mean value that preserves model utility.

### 4.3.2 IMPACT OF TASK VECTOR COMPONENTS

To validate the necessity of our two task vector approach, we conduct an ablation study examining each component's contribution (Table 5). The results reveal distinct and complementary roles. Without the $\Delta_1$, the model maintains high utility but achieves poor unlearning completeness, indicating that deleting collaborative signal alone fail to unlearn. Conversely, without $\Delta_2$, unlearning improves substantially but remains incomplete, demonstrating that collaborative patterns persist. The direct reconstruction case (w/o both)—which simply reconstructs from SVD without task vector operations—shows a marginal unlearning capability across all datasets. We attribute this slight improvement to the intrinsic properties of SVD; the decomposition of the user-item matrix into an orthogonal latent space effectively retains global collaborative patterns, thereby minimizing the impact of the removed data.

## 5 CONCLUSION

Recommendation unlearning fundamentally differs from conventional machine unlearning due to the entanglement of collaborative signals across the entire interaction space. In this paper, we propose **COVA** (**CO**llaborative **V**ector **A**rithmetic), a framework that combines the efficiency of task vector arithmetic with the collaborative nature of recommender systems. While task vectors traditionally operate through weight manipulation, we reinterpret them in the embedding space to align with collaborative filtering's learning mechanism. COVA develops this insight by leveraging SVD to construct a shared orthogonal space. Notably, since this space is established directly from matrices like prediction matrix, COVA inherently accounts for the collaborative patterns into the space. Our work demonstrates that preserving collaborative relationships while performing efficient unlearning is achievable through linear operations in the embedding space, opening new directions for recommendation unlearning.

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

## A   ANALYSIS OF ALTERNATIVE MATRIX CONFIGURATIONS

In this section, we analyze why joint decomposition of three matrices is necessary for effective recommendation unlearning, examining the limitations of alternative configurations.

**Single Matrix Approaches**   Using a single matrix fundamentally prevents task vector computation, as task vectors require computing differences between model states.

**Two-Matrix Configurations** We examine three possible two-matrix combinations and their inherent limitations:

- $(\mathbf{Y}_{\text{original}}, \mathbf{R}_{\text{pred}})$: This configuration captures the relationship between original interactions and model predictions. While it reveals training-induced transformations, it lacks information about the target unlearning state. Without $\mathbf{Y}_{\text{ideal}}$, we cannot specify which interactions should be removed or determine the desired end state, making unlearning impossible.

- $(\mathbf{Y}_{\text{ideal}}, \mathbf{R}_{\text{pred}})$: This pairing creates a fundamental misalignment. $\mathbf{R}_{\text{pred}}$ encodes collaborative patterns learned from $\mathbf{Y}_{\text{original}}$, including the interactions to be unlearned, while $\mathbf{Y}_{\text{ideal}}$ represents a state where these interactions never existed. This mismatch prevents proper capture of the unlearning transformation, as the matrices exist in conceptually incompatible spaces.

- $(\mathbf{Y}_{\text{original}}, \mathbf{Y}_{\text{ideal}})$: While this configuration clearly defines what needs to be removed, it lacks the model's learned representation. Without $\mathbf{R}_{\text{pred}}$, we have no access to the actual model requiring modification. The transformation exists only in data space, not in the trained model's embedding space where unlearning must occur.

The joint decomposition of all three matrices resolves these limitations by providing complete state representation ($\mathbf{Y}_{\text{original}}$ and $\mathbf{Y}_{\text{ideal}}$ define the transformation), model access ($\mathbf{R}_{\text{pred}}$ provides learned representations), and a unified coordinate system through COVA. This tripartite structure uniquely enables the computation of meaningful task vectors that capture both the unlearning direction and model-specific adjustments necessary for comprehensive unlearning.

## B ADDRESSING THE NEED OF HIGH-MEMORY RESOURCE

Applying Singular Value Decomposition (SVD) directly to the joint matrix $\tilde{\mathbf{A}} \in \mathbb{R}^{3|\mathbb{U}| \times |\mathbb{I}|}$ presents substantial computational challenges for large-scale recommender systems. Standard full SVD would necessitate constructing a $(3|\mathbb{U}|) \times (3|\mathbb{U}|)$ matrix with memory complexity of $O((3|\mathbb{U}|)^2)$, creating a prohibitive scalability bottleneck. To overcome this limitation, we employ randomized low-rank SVD, which computes accurate low-rank approximations without materializing the full square matrix. Hence, the computational complexity is reduced to $O(3|\mathbb{U}| \times k)$ from $O((3|\mathbb{U}|)^2)$ where $k$ represents the target rank with $k \ll 3|\mathbb{U}|$.

The procedure is outlined in Algorithm 1. Rather than directly decomposing the massive matrix $\tilde{\mathbf{A}}$, the algorithm constructs a low-dimensional sketch of the matrix's column space through an orthonormal basis $\mathbf{Q} \in \mathbb{R}^{3|\mathbb{U}| \times k}$. The basis serves as a compressed proxy that captures the most salient structural information of the joint user-item space. By projecting the full implicit matrix $\tilde{\mathbf{A}}$ onto the compact basis, we efficiently compute the low-rank approximation $\tilde{\mathbf{A}} \approx \mathbf{U}\boldsymbol{\Sigma}\mathbf{V}^\top$. Moreover, partitioning the left singular vectors $\mathbf{U}$ allows us to extract the user embeddings for each state—$\mathbf{U}_{\text{original}}$, $\mathbf{U}_{\text{ideal}}$, and $\mathbf{U}_{\text{pred}}$—which are central to our unlearning framework.

Despite the efficiency gains from randomized low-rank SVD, processing millions of users can still exceed GPU memory capacity since $\tilde{\mathbf{A}}$ must be materialized in GPU memory during computation. To address this constraint, we implement chunk-based processing that partitions $\tilde{\mathbf{A}}$ into $c$ segments of size $\frac{3|\mathbb{U}|}{c} \times n$, thereby avoiding the need to store the entire matrix in GPU memory simultaneously. Matrix operations such as the sketch computation $\mathbf{X} = \tilde{\mathbf{A}}\boldsymbol{\Omega}$ are decomposed as:

$$\mathbf{X} = \tilde{\mathbf{A}}\boldsymbol{\Omega} = \begin{bmatrix} \tilde{\mathbf{A}}^{(1)} \\ \tilde{\mathbf{A}}^{(2)} \\ \vdots \\ \tilde{\mathbf{A}}^{(c)} \end{bmatrix} \boldsymbol{\Omega} = \begin{bmatrix} \tilde{\mathbf{A}}^{(1)}\boldsymbol{\Omega} \\ \tilde{\mathbf{A}}^{(2)}\boldsymbol{\Omega} \\ \vdots \\ \tilde{\mathbf{A}}^{(c)}\boldsymbol{\Omega} \end{bmatrix}, \tag{8}$$

where each chunk $\tilde{\mathbf{A}}^{(i)} \in \mathbb{R}^{\frac{3|\mathbb{U}|}{c} \times n}$ is processed independently, reducing peak GPU memory requirements from $O(3|\mathbb{U}|n)$ to $O(\frac{3|\mathbb{U}|n}{c})$. The chunking strategy enables processing of datasets that exceed available GPU memory while preserving computational efficiency. Although sequential processing introduces some overhead, empirical results demonstrate that our implementation maintains significant speedup compared to baseline, validating the favorable memory-computation trade-off for large-scale recommendation unlearning.

---

**Algorithm 1** Randomized Low-Rank SVD

---

**Require:** $\tilde{\mathbf{Y}}_{\text{original}}, \tilde{\mathbf{Y}}_{\text{ideal}}, \mathbf{R}_{\text{pred}}$, target rank $k$, the number of iterations $p$.
**Ensure:** $\mathbf{U}_{\text{original}}, \mathbf{U}_{\text{ideal}}, \mathbf{U}_{\text{pred}} \in \mathbb{R}^{|\mathbb{U}| \times k}, \boldsymbol{\Sigma} \in \mathbb{R}^{k \times k}, \mathbf{V} \in \mathbb{R}^{|\mathbb{I}| \times k}$.

1: Define the joint matrix: $\tilde{\mathbf{A}} \leftarrow \begin{bmatrix} \tilde{\mathbf{Y}}_{\text{original}} \\ \tilde{\mathbf{Y}}_{\text{ideal}} \\ \mathbf{R}_{\text{pred}} \end{bmatrix}$.

    **Stage A: Find an approximate basis for the range of $\tilde{\mathbf{A}}$**
2: Draw a random Gaussian matrix $\boldsymbol{\Omega} \in \mathbb{R}^{|\mathbb{I}| \times k}$.
3: Form the matrix sketch $\mathbf{X} \leftarrow \tilde{\mathbf{A}}\boldsymbol{\Omega}$.
4: Compute an orthonormal basis via QR factorization: $[\mathbf{Q}, \sim] \leftarrow \text{qr}(\mathbf{X})$.
5: **for** $j = 1, \ldots, p$ **do**               ▷ Power iterations to refine the basis
6:     $\mathbf{X}' \leftarrow \tilde{\mathbf{A}}^{\top}\mathbf{Q}$
7:     $[\mathbf{Q}', \sim] \leftarrow \text{qr}(\mathbf{X}')$
8:     $\mathbf{X} \leftarrow \tilde{\mathbf{A}}\mathbf{Q}'$
9:     $[\mathbf{Q}, \sim] \leftarrow \text{qr}(\mathbf{X})$
10: **end for**
    **Stage B: Form the low-rank SVD using the basis Q**
11: Form the small projected matrix $\mathbf{B} \leftarrow \mathbf{Q}^{\top}\tilde{\mathbf{A}}$.
12: Compute the SVD of the small matrix: $[\tilde{\mathbf{U}}, \boldsymbol{\Sigma}, \mathbf{V}^{\top}] \leftarrow \text{svd}(\mathbf{B})$.
13: Form the final left singular vectors: $\mathbf{U} \leftarrow \mathbf{Q}\tilde{\mathbf{U}}$.
14: Partition $\mathbf{U} \in \mathbb{R}^{3|\mathbb{U}| \times k}$ into three $|\mathbb{U}| \times k$ blocks:
15: $\begin{bmatrix} \mathbf{U}_{\text{original}} \\ \mathbf{U}_{\text{ideal}} \\ \mathbf{U}_{\text{pred}} \end{bmatrix} \leftarrow \mathbf{U}$
16: **return** $\mathbf{U}_{\text{original}}, \mathbf{U}_{\text{ideal}}, \mathbf{U}_{\text{pred}}, \boldsymbol{\Sigma}, \mathbf{V}$.

---

## C   DETAILS SETTINGS

### C.1   PREPROCESSING DETAILS

We conduct experiments on Yelp2018 (He et al., 2020), Gowalla (Cho et al., 2011), and Amazon-Book (Ni et al., 2019), three standard benchmarks commonly adopted in CF research with dataset statistics in Table 6. For implicit feedback conversion, we binarize all ratings using a threshold of

Table 6: **Statistics of the datasets.**

| Dataset | # Users | # Items | # Interactions | Density |
|---|---|---|---|---|
| Yelp2018 | 31,831 | 40,841 | 1,666,869 | 0.013% |
| Gowalla | 29,858 | 40,981 | 1,027,370 | 0.084% |
| Amazon-Book | 52,643 | 91,599 | 2,984,108 | 0.062% |

4, where ratings above this threshold are converted to positive labels and others are treated as unobserved interactions. This transformation aligns with the implicit feedback paradigm commonly adopted in CF literature (Hu et al., 2008; Ostuni et al., 2013; Zhou et al., 2018). To ensure data quality, we apply 5-core filtering that retains only users and items with at least five interactions (He et al., 2020). This process reduces data sparsity while maintaining sufficient density for meaningful collaborative signals. For unlearning setup, we randomly sample a subset of training interactions as the forget set, with the complement constituting the retain set (Zhang et al., 2024). Following practical deployment considerations where unlearning requests typically involve minimal data relative to the training corpus, we designate 1% of training interactions as the default deletion ratio for unlearning. We further investigate robustness against different ratios in Section 4.2. All preprocessing uses fixed random seeds for reproducibility across baselines and our method. The data is split into training (60%), validation (20%), and test (20%) sets following standard protocols (Zhang et al., 2024).

### C.2   DETAILS OF BASELINES

We provide detailed descriptions of the baseline methods used for comparison in our experiments. SISA (Bourtoule et al., 2021) is a foundational partition-based unlearning framework that operates by sharding the training data into multiple disjoint sets and training an independent sub-model on

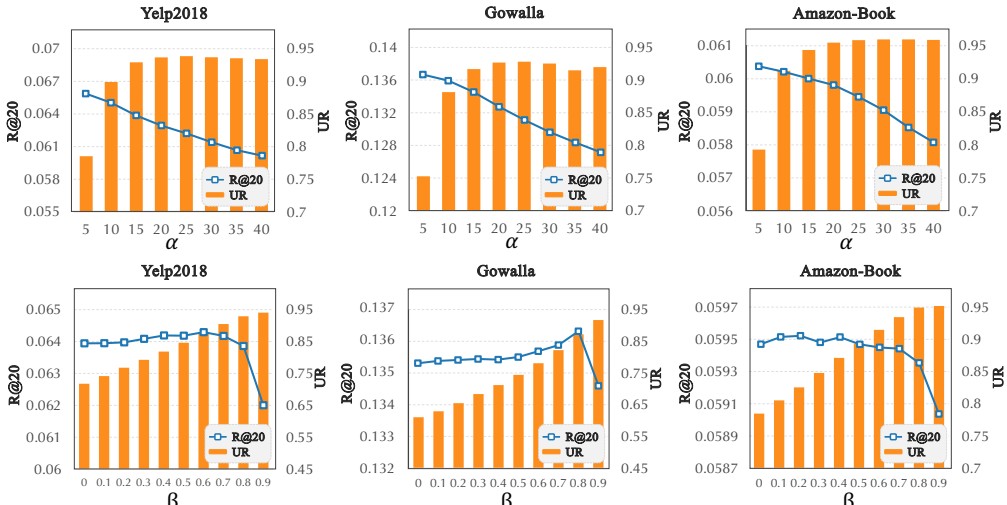

Figure 3: **Parameter sensitivity with regard to hyperparameters $\alpha$ and $\beta$** An increase in $\alpha$ positively correlates with unlearning performance (UR), but this comes at the cost of a consistent decline in recommendation quality (R@20). In contrast, increasing $\beta$ shows a tendency to improve R@20.

each; when an unlearning request is received, only the sub-models trained on shards containing the data to be forgotten are retrained. RecEraser (Chen et al., 2022) adapts this sharding architecture for collaborative filtering by leveraging user embedding clustering to create more homogeneous data shards to mitigate the impact of retraining on users with similar preferences. SCIF (Li et al., 2023b) utilizes influence functions to approximate the effect of data removal on model parameters, enabling efficient unlearning by selectively updating only the most influenced parameters. Lastly, IFRU (Zhang et al., 2024) is an influence function-based method tailored for graph-based recommender systems that extends conventional influence calculations by explicitly modeling the spillover effects of unlearning an interaction, which can propagate through the user-item interaction graph.

### C.3 Hyperparameter Analysis

We conduct a sensitivity analysis on the hyperparameters $\alpha$ and $\beta$ by varying their values to understand their impact on both unlearning performance and recommendation quality by Unlearning Ratio (UR), Recall@20 (R@20), respectively. As shown in Figure 3, our experiments confirm that $\alpha$ and $\beta$ maintain these consistent roles and trends across all three datasets. We use the $\alpha$ parameter to directly control the strength of the unlearning vector, $\Delta_1$. As the value of $\alpha$ increases, UR consistently rises, indicating a more aggressive removal of target interactions. However, this leads to a greater loss of useful context that ought to be preserved, resulting in degraded R@20. In parallel, $\beta$ controls the influence of the learned pattern vector, $\Delta_2$. When $\beta$ increases, UR also consistently rises, as it helps to remove latent traces that $\Delta_1$ alone cannot address. Furthermore, the stability or slight improvement in R@20 suggests that the $\beta$ corrects for noise accumulated during the optimization process. However, R@20 drops sharply after $\beta$ exceeds a certain threshold, mirroring the behavior observed with $\alpha$. This indicates that an overly aggressive unlearning of collaborative effects can also inadvertently discard valuable knowledge. To sum up, we conclude that $\alpha$ primary affects unlearning performance, while $\beta$ serves as regulator that both enhances unlearning and mitigates optimization noise.

### C.4 Detailed Experimental Setup

All experiments are conducted on a single NVIDIA RTX A6000 GPU. Our experimental settings are adapted from the IFRU implementation (Zhang et al., 2024). All baseline models are trained for a maximum of 5000 epochs with a batch size of 2048, and model embeddings are initialized from a Gaussian distribution $\mathcal{N}(0, 0.01)$. We employ an early stopping strategy based on Recall@20 on the

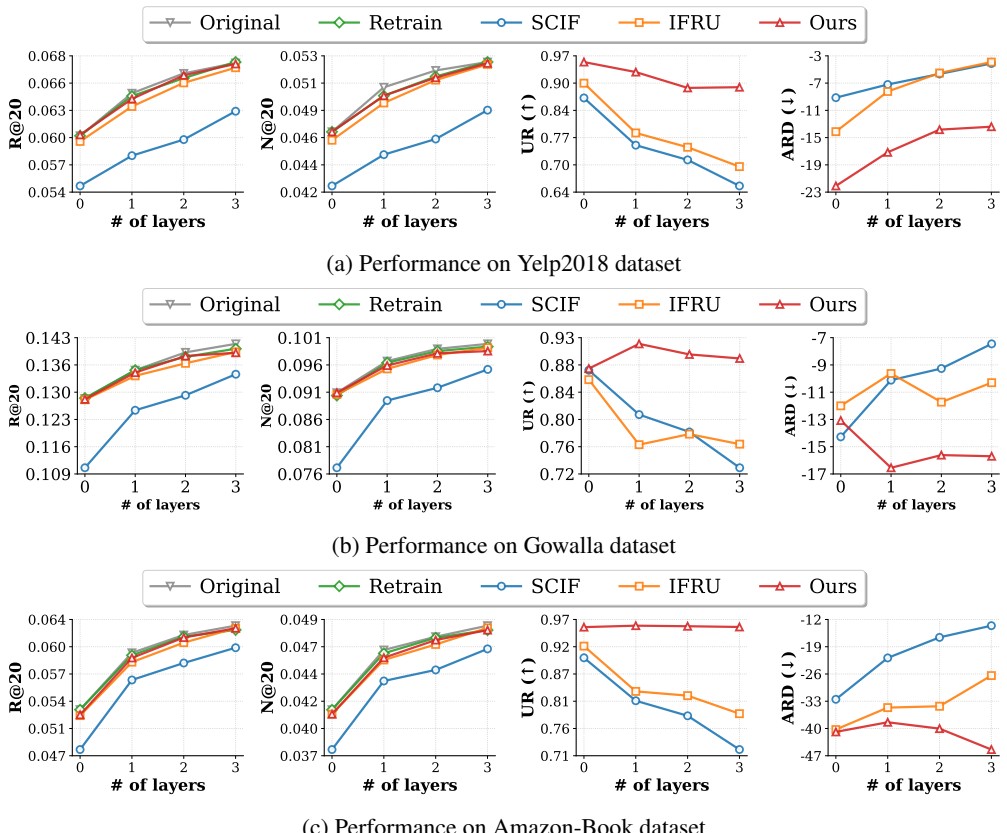

Figure 4: **Unlearning performance across datasets with varying layer depths.** While deeper layers enhance recommendation utility, existing methods suffer from increased user-item entanglement, whereas our method maintains consistent unlearning effectiveness.

validation set with a patience of 10 epochs. To prevent significant degradation of recommendation utility, we enforce an additional termination condition for iterative unlearning methods; training is stopped if Recall@20 falls below 90% of the existing model's performance. In particular, this condition does not apply to SISA (Bourtoule et al., 2021) and RecEraser (Chen et al., 2022) since they retrain affected partitions.

For the training objective, we employ the Bayesian Personalized Ranking (BPR) loss (Rendle et al., 2009) for all models to better align with the inherent ranking nature of recommendation tasks. This choice represents a divergence from the Binary Cross-Entropy (BCE) loss used in prior works (Chen et al., 2022; Li et al., 2023b; Zhang et al., 2024). To facilitate effective training with BPR loss, we utilize negative sampling. The sampling strategy is adapted to the model architecture; for partition-based models, negative samples are drawn exclusively from within the same data partition to maintain its integrity, while standard uniform negative sampling is used for all other methods.

Our proposed model, COVA, is a non-iterative algorithm, thus early stopping is not applicable. Instead, we select the optimal configuration by identifying all hyperparameter settings that maintain at least 95% of the original model's Recall@20, and from this subset, we choose the model that achieves the highest unlearning ratio. This strategy ensures both maximal unlearning completeness and preserving recommendation utility. For the low-rank SVD, the rank $k$ is set to 48, aligning with the embedding dimension of the base model. We fix the number of power iterations for the randomized SVD at 10. While the algorithm is stochastic, it exhibits high stability, and we use a fixed random seed in all experiments to ensure full reproducibility. This choice is based on our empirical observation that tuning this value does not reveal a consistent pattern of improvement. Our main performance comparisons are conducted using a single chunk across all datasets.

## D    ANALYSIS OF COLLABORATIVE EFFECTS

In this section, we provide a comprehensive analysis of how GCN layer depth, which modulates the strength of collaborative signals, impacts unlearning performance. The full results across all datasets are presented in Figure 4. The primary observation is the trade-off between recommendation accuracy and unlearning completeness in baseline methods. As depicted, increasing the number of layers from 0 to 3 universally boosts recommendation metrics (R@20, N@20). However, for SCIF and IFRU, this comes at the cost of a severe decline in Unlearning ratio (UR). This suggests that the enriched collaborative signals in deeper GCNs create complex entanglements that these methods cannot effectively resolve during the unlearning process. In contrast, COVA demonstrates immunity to this negative trade-off. Its UR remains consistently high regardless of network depth. More notably, on datasets such as Amazon-Book, the ranking deterioration (ARD) measure intensifies with deeper layers. This counter-intuitive finding indicates that COVA is not only robust to stronger collaborative signals but can actually repurpose them to enhance the unlearning process, turning a challenge for other methods into an advantage. This unique characteristic underscores COVA's suitability for modern deep GCN architectures where unlearning is a critical requirement.

## E    UNDERSTANDING COLLABORATIVE SIGNAL DISTRIBUTION IN TASK VECTORS

The key insight behind our two-component task vector design lies in recognizing the distinct information encoded within each matrix state after SVD mapping. When decomposed into the shared orthogonal space through SVD, the embedding matrices $\mathbf{U}_{\text{ideal}}$, $\mathbf{U}_{\text{original}}$, and $\mathbf{U}_{\text{pred}}$ each primarily capture different aspects of the interaction landscape. The $\mathbf{U}_{\text{ideal}}$ embeddings mainly represent the interaction patterns that should remain after unlearning, containing ideal state information. The $\mathbf{U}_{\text{original}}$ embeddings include both this ideal information and explicit information about the interactions to be unlearned, representing the complete dataset structure mapped into the latent space. The $\mathbf{U}_{\text{pred}}$ embeddings, however, contain the most comprehensive representation—they encode both types of explicit information (ideal state and target interactions) plus, crucially, the collaborative information that emerges from the model's optimization process. This collaborative component represents the learned associations and implicit patterns that CF models extract during training, extending far beyond the direct interaction structure.

Given this decomposition, our design of $\Delta_1 = \mathbf{U}_{\text{original}} - \mathbf{U}_{\text{ideal}}$ naturally captures the explicit structural difference—the direct presence of target interactions in the data. However, this alone cannot achieve complete unlearning because CF's optimization distributes each interaction's influence throughout the embedding space via collaborative learning. The second component $\Delta_2 = \mathbf{U}_{\text{pred}} - \mathbf{U}_{\text{original}}$ specifically targets these learned collaborative patterns by isolating the transformation from raw interaction structure to trained model state. Through this separation, we can remove both the direct structural footprint of deleted interactions (via $\Delta_1$) and their distributed collaborative influence (via $\Delta_2$), enabling comprehensive unlearning that accounts for CF's unique propagation of interaction effects across the entire latent space.

## F    LIMITATION

While COVA demonstrates significant improvements in unlearning effectiveness and efficiency, we acknowledge certain limitations in our current framework. First, the normalization strategy we employ, though empirically validated through ablation studies in Section 4.3.1, may not constitute the optimal approach across all dataset characteristics or model architectures. We recognize that more advanced normalization techniques could potentially enhance performance; however, our research primarily focuses on establishing a novel unlearning paradigm for collaborative filtering rather than exhaustively optimizing normalization methodologies. The current design serves as a sufficient proof-of-concept that enables the core contribution of embedding-space task vector arithmetic.

Second, our choice of SVD as the decomposition method, while theoretically grounded and empirically successful, represents one of several possible approaches for constructing shared orthogonal spaces. Alternative decomposition techniques such as other decomposition methods could potentially offer different trade-offs between computational efficiency and representation quality. We

select SVD for its well-established theoretical properties and computational maturity, but acknowledge that other methods might better capture specific aspects of collaborative patterns. This choice reflects our prioritization of establishing the feasibility of embedding-space task vector arithmetic rather than exhaustively comparing all possible decomposition strategies.

