# OpenReview forum: "Efficient Recommendation Unlearning via Task Vector Arithmetic in Shared Space"
_ICLR.cc/2026/Conference — ICLR 2026 Conference Withdrawn Submission_

### Official Review · Reviewer_4qf6 · 2025-10-27

**Soundness:** 2
**Presentation:** 3
**Contribution:** 2
**Rating:** 4
**Confidence:** 4

**Summary:**

This paper focuses on the problem of machine unlearning for recommendation systems. Two of the core challenges that unlearning faces in recommendation system settings are (a) fragmentation of the interaction structure, coming from partition-based methods, and (b) expensive optimization processes, coming from influence-based methods. To address these concerns, the authors propose COVA, a method which aims to remove the influence of interactions by learning task vectors that one can perform arithmetic over. To do this, SVD is performed over a matrix comprised of the true user-item interaction set, the user-item interaction set with unlearned interactions removed, and the final prediction matrix. Then, the result of the SVD is used to remove both the presence of the interactions to remove, as well as their influence that comes from the CF training algorithm. The authors show strong results on a series of benchmark datasets with both MF and LightGCN backbones with significant efficiency gains for unlearning.

**Strengths:**

I think this work offers a simple, yet effective method to perform unlearning on CF models. Some strengths include:

1. The performance vs efficiency trade-off is very strong. The authors are able to retain highly competitive performance as compared to the base model without much computational overhead. I appreciate the amount of experiments and rigor that went in to characterizing the computational efficiency of the method.
2. The paper is generally easy to follow, although this is largely from the simplicity of the method.
3. I thought the unlearning metrics for the CF models were a nice addition that could be beneficial for work in this space.
4. I think the connection to graph unlearning is meaningful and helpful to contextualize some of the earlier challenges acknowledged by the work in unlearning for recommender systems.

**Weaknesses:**

Despite the empirical strengths of the paper, I have a few concerns with the generalization and practical utility of the work.

1. Linear Models: The authors only study MF and LightGCN, models that are linear with respect to the user and item embeddings. It seems the core reason COVA is so fast is that it exploits this property. That said, if one were to use non-linear methods (NeuMF, NGCF, etc.), it is unclear what would happen. Moreover, the methods that the authors compare against do not necessarily depend on the linear relationship, and thus one concern could be that these methods would perform better in the more general, non-linear setting.
2. Theoretical Analysis: The most obvious weakness of the paper is lack of theoretical justification for the method. While theory is not always a necessity, the whole paper hinges on linear properties that can be studied rigorously. Also, given the implications of unlearning in practice, it is important to offer some guarantees or theoretical justification for why this method should be adopted. As of now, the design of the method is largely driven by intuition, rather than rigorous justification. I also think this is an area to increase the novelty of the work given the majority of the method uses components that already exist.
3. Hyperparams: Given the relative simplicity of the method, it would be nice to have a deeper study on alpha and beta and how they interplay with one another. In the appendix, the authors claim insensitivity, but, to me, the plots tell a different story. While there are regions of the stability, tuning alpha or beta seem to produce a direct trade-off in performance and unlearning, and there are is little direct guidance from the authors on how to tune these in practice. I also think it would be nice to jointly show how alpha and beta interact with one another.
4. Negative Sampling and Losses: While the authors use BPR to train their models, presumably with a single negative sample, they make no discussion on how negative sampling could impact their method. For instance, negative sampling can create influence that is not explicitly captured in the interaction matrix. Moreover, BPR (and its one negative sample) is known to not perform as well compared to methods which use more negative samples, like sampled softmax, which can further hurt COVA as the interaction patterns during computation become much denser.

**Questions:**

While I have elaborated on my questions in the weaknesses section, below are more concise versions of these questions:

1. How does COVA perform with deeper, non-linear models, particularly when compared to the other methods which are not as tightly coupled to linear properties.
2. Can any guarantees or more rigorous justification be provided for COVA?
3. How do alpha and beta interact with one another? How difficult are they to jointly tune?
4. What happens when negative sampling is considered, both in terms of the motivation/justification or COVA and empirical results?

---

### Official Review · Reviewer_wxdw · 2025-10-29

**Soundness:** 2
**Presentation:** 3
**Contribution:** 2
**Rating:** 2
**Confidence:** 4

**Summary:**

This paper proposes COVA, a novel framework for efficient recommendation unlearning that addresses the limitations of partition-based and influence-function-based approaches. Unlike previous methods that either disrupt collaborative patterns or rely on iterative parameter updates, COVA performs unlearning directly in the embedding space using task vector arithmetic within a shared orthogonal latent space constructed via singular value decomposition (SVD). This design allows COVA to remove both structural and collaborative influences of deleted interactions from the model’s prediction matrix without retraining or accessing model internals.

**Strengths:**

1. The problem addressed is practical, well-motivated, and important for privacy-preserving recommender systems.
2. The idea of adapting task vector arithmetic, previously used in NLP/CV model editing, to the embedding space of collaborative filtering is conceptually interesting and innovative.
3. The output-level unlearning approach — operating directly on the prediction matrix rather than modifying model parameters — enhances applicability and deployment feasibility, as it avoids retraining or model access.
4. The paper is clearly written and well-structured, making it easy to follow.

**Weaknesses:**

1. In line 43, the authors claim that “unlike NLP or CV, data instances are independent”, which is misleading. In fact, NLP and CV data also exhibit shared feature subspaces or correlated representations, meaning that unlearning one instance can induce global influence.
2. The paper repeatedly distinguishes “embeddings” in CF models from “parameters” in other deep models. However, embeddings are indeed model parameters, merely representing a different form of parameterization. Hence, the claim that this work moves task vector arithmetic from parameter space to embedding space is conceptually weak.
3. In Section 3.1, the authors concatenate three matrices and apply SVD to construct a shared latent space. However, decomposing this vertically concatenated matrix only guarantees that all resulted vectors of shape 3|U| are in a shared space and does not guarantee that |U|-sized partitions share a coherent space for subsequent arithmetic operations. The mathematical validity of performing task vector operations on these split embeddings thus requires further justification.
4. Although the paper motivates its contribution in the context of data privacy, COVA operates solely on the prediction matrix while keeping model parameters unchanged. Thus, the underlying data information may remain encoded in the model weights. From this perspective, comparing COVA to parameter-modifying unlearning methods may be unfair, as the latter genuinely reduce privacy leakage risk. If privacy protection is claimed, the paper should report privacy leakage probabilities (via differential privacy or membership inference analysis).
5. The proposed metrics, UR and ARD, appear reasonable based on the released code, but without a gold standard reference value, it is unclear how to interpret the reported performance or determine whether it achieves satisfactory unlearning completeness.

**Questions:**

1. In Section 3.1, why do the authors normalize $$Y_{\text{original}}$$ and $$Y_{\text{ideal}}$$ but not $$R_{\text{pred}}$$?
2. Why is it theoretically valid to assume that the three |U|-dimensional partitions of the decomposed matrix reside in the same shared space?
3. What is the gold standard reference for the two proposed metrics (UR and ARD)?
4. Have the authors evaluated ranking persistence on non-forgotten interactions to confirm that unlearning does not degrade unaffected recommendations? (a metric similar to UR and ARD)
5. A conceptual paradox arises: if “retraining” is considered the gold standard and the proposed metrics are valid, COVA’s performance should not exceed retraining. However, Table 1 shows that COVA outperforms retraining in several cases. This discrepancy suggests one of two possibilities: (1) Retraining may not be an appropriate gold standard, in which case approximating retraining should no longer be framed as the methodological goal, and $$Y_{\text{ideal}}$$ may not serve as a valid reference state; or (2) the proposed metrics may not faithfully capture true unlearning behavior. The authors are encouraged to clarify which interpretation they support and how this affects the conceptual positioning of their work.
6. The paper should explicitly present the mathematical definitions of UR and ARD in the main text, rather than referring readers to the anonymized code.
7. Please also address the concerns raised in the Weaknesses section.

---

### Official Review · Reviewer_ME5r · 2025-11-01

**Soundness:** 3
**Presentation:** 3
**Contribution:** 2
**Rating:** 4
**Confidence:** 4

**Summary:**

This paper addresses the challenge of achieving efficient and privacy-preserving unlearning in recommender systems and introduces a novel framework named COVA (Collaborative Vector Arithmetic). COVA reinterprets task vector arithmetic from parameter space to the embedding space, constructing a shared orthogonal latent space via SVD on the prediction matrix. Unlearning is then achieved by subtracting task vectors, enabling output-level modification without model retraining. The authors validate the effectiveness of their method through extensive experiments.

**Strengths:**

1. The method is elegantly designed, combining singular value decomposition (SVD) with task vector arithmetic to preserve collaborative structures while enabling efficient and rapid unlearning.
2. The experimental evaluation is comprehensive, employing the newly proposed UR and ARD metrics that better align with the characteristics and goals of recommender systems.
3. The paper is clearly written and generally well organized.

**Weaknesses:**

1. Evaluation is limited to ranking-based metrics, lacking probabilistic measures of unlearning completeness.
2. The claim that COVA outperforms retraining lacks clear explanation or evidence.
3. Performance is highly sensitive to normalization, showing limited robustness across datasets.

**Questions:**

1. The evaluation currently relies only on ranking-based metrics, which cannot reflect changes in model confidence or the true extent of forgetting. Could the authors include score-based or membership inference evaluations (MIA) to confirm whether the model genuinely forgets target data at the probabilistic level?
2. The unlearning set is randomly sampled. Have the authors examined targeted unlearning cases—such as removing specific users or items—to test robustness and real-world applicability?
3. The paper attributes COVA’s performance surpassing retraining and even the original model to the Δ₂ term acting as a “collaborative noise removal” component. Could the authors provide further theoretical justification or visual evidence to support this claim?

---

### Official Review · Reviewer_VzJv · 2025-11-01

**Soundness:** 1
**Presentation:** 3
**Contribution:** 2
**Rating:** 2
**Confidence:** 4

**Summary:**

This paper focuses on recommendation unlearning. The authors propose an output-level unlearning approach based on task vector arithmetic. The authors further employ SVD to construct a unified embedding space to facilitate the application of task vector arithmetic. Experiments are conducted on multiple real-world datasets.

**Strengths:**

1. The authors identify the key issues of existing approaches, e.g., P-based and IF-based. Note that there is also fine-tuning based approach.
2. Introducing task vector arithmetic provides new insights into recommendation unlearning research.
3. The writing is clear and easy to follow.

**Weaknesses:**

1. This study focuses on output-level unlearning, which is a limiting setting.‌ It is neither applicable in black-box scenarios nor useful against attackers in white-box settings.
2. The results show that COVA achieves better completeness than retraining, which can be interpreted as a sign of over-unlearning.‌ The ablation study also reveals that removing delta2 (interpreted as collaborative information to be removed during unlearning) leads to more over-unlearning, which is the opposite of the intended interpretation of delta2. This further highlights a critical weakness of the paper: the key definition of vector subtraction (delta1, delta2) is purely intuitive, lacks theoretical support, and conflicts with empirical evidence.
3. Figure 2 shows that COVA's advantage diminishes as the unlearning ratio increases.‌ Can COVA guarantee its advantage with large unlearning ratios? Note that for small ratios (1% and 0.1%), existing unlearning approaches can be highly efficient. I do not agree that using COVA+chunk in terms of efficiency comparison is fair.
4. An important baseline is missing for comparison: UltraRE.‌
5. Minor:‌ Using random samples as the forget set cannot effectively evaluate unlearning performance, especially in recommendation unlearning [1]. Testing across different sample levels would be more insightful.

[1] CURE4Rec: A Benchmark for Recommendation Unlearning with Deeper Influence. NeurIPS’24

**Questions:**

Please see weaknesses.

---

### Note · Authors · 2025-11-28

I have read and agree with the venue's withdrawal policy on behalf of myself and my co-authors.